# A Parametric Study of Flushing Conditions for Improvement of Angioscopy Visibility

**DOI:** 10.3390/jfb13020069

**Published:** 2022-06-01

**Authors:** Kohei Mitsuzuka, Yujie Li, Toshio Nakayama, Hitomi Anzai, Daisuke Goanno, Simon Tupin, Mingzi Zhang, Haoran Wang, Kazunori Horie, Makoto Ohta

**Affiliations:** 1Institute of Fluid Science, Tohoku University, 2-1-1 Katahira, Aoba-ku, Sendai 980-8577, Japan; k.mitsuzuka0326@gmail.com (K.M.); jessie.li@torrens.edu.au (Y.L.); hitomi.anzai.b5@tohoku.ac.jp (H.A.); daisuke.goanno.r1@dc.tohoku.ac.jp (D.G.); s.tupin@tohoku.ac.jp (S.T.); mingzi.zhang@mq.edu.au (M.Z.); victorytcwang@gmail.com (H.W.); 2Graduate School of Biomedical Engineering, Tohoku University, 6-6 Azaaoba, Aramaki, Aoba-ku, Sendai 980-8579, Japan; 3Centre for Healthy Futures, Torrens University Australia, 1-51 Foveaux Street, Sydney, NSW 2010, Australia; 4National Institute of Technology, Nara College, 22 Yatacho, Yamatokoriyama 639-1080, Japan; tnakayama@ctrl.nara-k.ac.jp; 5Macquarie Medical School, Faculty of Medicine, Health, and Human Sciences, Macquarie University, 75 Talavera Rd., Sydney, NSW 2109, Australia; 6Department of Cardiology, Sendai Kousei Hospital, 4-15 Hirose, Aoba-ku, Sendai 980-0873, Japan; horihori1015@gmail.com

**Keywords:** coronary angioscopy, flush conditions, CFD, two-phase flow, dextran injection

## Abstract

During an angioscopy operation, a transparent liquid called dextran is sprayed out from a catheter to flush the blood away from the space between the camera and target. Medical doctors usually inject dextran at a constant flow rate. However, they often cannot obtain clear angioscopy visibility because the flushing out of the blood is insufficient. Good flushing conditions producing clear angioscopy visibility will increase the rate of success of angioscopy operations. This study aimed to determine a way to improve the clarity for angioscopy under different values for the parameters of the injection waveform, endoscope position, and catheter angle. We also determined the effect of a stepwise waveform for injecting the dextran only during systole while synchronizing the waveform to the cardiac cycle. To evaluate the visibility of the blood-vessel walls, we performed a computational fluid dynamics (CFD) simulation and calculated the visible area ratio (*VAR*), representing the ratio of the visible wall area to the total area of the wall at each point in time. Additionally, the normalized integration of the *VAR* called the area ratio (***AR_VAR_***) represents the ratio of the visible wall area as a function of the dextran injection period. The results demonstrate that the ***AR_VAR_*** with a stepped waveform, bottom endoscope, and three-degree-angle catheter results in the highest visibility, around 25 times larger than that under the control conditions: a constant waveform, a center endoscope, and 0 degrees. This set of conditions can improve angioscopy visibility.

## 1. Introduction

Ischemic heart disease (IHD) (e.g., heart infarction and angina pectoris) has been a primary cause of death in the world [1]. The IHD is caused by stenosis inside the coronary artery [2]. With the chronic deposition of cholesterols and other substances in the blood, the arterial lumen becomes narrow, and the blood flow is partially or totally blocked. The lack of oxygen leads to the dysfunction of heart muscle. When medical doctors perform percutaneous coronary intervention (PCI) as a treatment method for IHD, they need to evaluate the condition of the thrombus and plaque before and after the procedure to perform the PCI procedure safely. The color of the thrombus and plaque is an essential marker of their condition [3,4].

Angiography, intravascular ultrasound (IVUS) [5,6,7] and optical coherence tomography [8,9] are frequently used to observe the blood-vessel wall. However, they provide only an indirect view of the plaque. Angioscopy provides sequential visual images through a camera and directly visualizes the wall and plaque. Medical doctors seek color information for the plaque on the wall, thrombus, and plaque protrusion, and the vessel’s morphology [10,11]. There are several papers with statistical analyses describing the capabilities of angioscopy for plaque-color classification or stent-coverage investigations [12,13]. During angioscopy, for taking images, the blood in the artery needs to be removed by using a transparent liquid. Low-molecular-weight dextran is often used as the transparent liquid [14,15]. Previously, the coronary artery was completely blocked using a balloon to stop the blood flow. Then, dextran was sprayed out of the catheter to remove the blood. This method allowed doctors to record images easily [16]. However, balloon occlusion poses a risk of IHD because of the complete cessation of the blood supply [17], so the use of a balloon is gradually being phased out to avoid this risk [18]. The new method is performed without a balloon, and the dextran is sprayed out while the images are being taken. Komatsu et al. examined the possibility of increasing the visibility by increasing the flow rate [17]. However, there is still a risk of blood-flow blockage due to over-flushing.

To improve the visibility during angioscopy, several studies on shape improvements for the catheter have been conducted. Li et al. proposed several optimized designs for the catheter [19]. Yamakoshi et al. developed a catheter with a hood geometry at its tip for keeping the dextran in front of the camera [20]. Okayama et al. developed a catheter with holes on its side surface to spray the dextran to the wall, to make the flush flow reach the vessel wall [21]. Additionally, Faisal et al. investigated the flushing flow conditions and pressure inside the artery during angioscopy using a simplified model to optimize the hole shape on the catheter [22]. These studies and developments appear promising but realizing the use of these new devices in treatment may take time, as their effectivity is verified.

We hypothesize that the visibility can be improved not only with one condition, but with several conditions. Several candidate conditions that could affect the angioscopy visibility are shown in Table 1. Some of these may have more substantial effects on visibility. The improvement of these conditions will be applicable to not only the conventional devices, but also the next generation of devices. Therefore, this study sought to determine a method for realizing clearer angioscopy under different values for the parameters of the injection waveform, endoscope position, and catheter angle, using computational fluid dynamics (CFD).

## 2. Materials and Methods

### 2.1. Model

Figure 1 shows coronary vascular, catheter, and endoscope models. These models were constructed using CAD software (SolidWorks, Dassault, France). The coronary vascular model was a straight 2.00 mm-diameter vessel. The angioscopy model consisted of a straight catheter and an endoscope. The straight catheter was set in the center of the blood vessel. The outer and inner diameters of the catheter were 1.70 and 1.42 mm, respectively. The total lengths of the blood vessel and catheter were 35.00 and 7.55 mm, respectively. The endoscope model (Smart-iTM type S11, SURGE TEC Corp., Tokyo, Japan) consisted of the camera, 0.50 mm-diameter camera cord, and 0.36 mm-diameter guidewire model. The shape of the camera featured a lean on the camera lens. The camera lens was assumed to be placed at the orange point shown in Figure 2. The center axis of the camera lens was 0.30 mm away from the blood vessel’s central axis in the y-direction. The viewing angle of the camera was set at 60 degrees based on the instructions for the endoscope. The endoscope was set in the center of the catheter, and this model was called the center model. In this study, the depth of the field was set as unlimited.

Here, we changed the endoscope position and catheter angle with respect to the angioscope geometry. Three models were prepared to change the endoscope position in the catheter. The endoscope was brought into contact with the top, side, and bottom wall of the catheter, respectively (Figure 3A). Additionally, the models with the top, side, and bottom endoscopes were called the top, side, and bottom (0°) models, respectively. Three models with different catheter angles were prepared. The 2.55 mm-length part of the catheter tip was angled at 1, 2, and 3 degrees in the bottom direction (Figure 3B). The endoscope in the angled catheter model was brought into contact with the catheter’s bottom inner wall. The models with the 1-, 2-, and 3-degree catheters were called the bottom (1°), bottom (2°), and bottom (3°) models, respectively.

The center model was used to study the effect of the injection waveform on the angioscopy visibility. Additionally, the stepped waveform was applied on all the other cases.

The fluid domains of the blood vessel and catheter were discretized into tetrahedron elements using ANSYS Meshing 2019 R1 (ANSYS Inc., Canonsburg, PA, USA), with finer elements of less than 0.0860 mm for the blood-vessel domain and 0.0220 and 0.0710 mm for the domain around and before the camera in the catheter, respectively. These mesh sizes were decided on based on the results of the mesh sensitivity analysis according to the grid convergence index (GCI) [23]. The mesh numbers of each model are shown in Table 2.

### 2.2. Boundary Condition

As the blood flow rate at the inlet, the physiological pulsatile blood flow rate of the left anterior descending artery shown in Figure 4 was applied to a previous study [24]. The maximum and minimum blood flow rates were 54 and 9 mL/min, respectively. Two patterns of a dextran boundary condition with different injection waveforms were compared (Figure 4). Figure 4a shows the boundary condition for reproducing a situation in which a doctor injected dextran continuously under a constant-flow-rate condition within the dextran volume limitation. This injection waveform is called a constant waveform. Figure 4b shows the boundary condition for injecting dextran only during a 0.5 s systole. In this condition, it became possible to increase the maximum dextran flow rate up to 180 mL/min while the total volume of the dextran was maintained. This injection waveform is called the stepped waveform. The cardiac cycle of both the blood and dextran flow rate curve was 1.0 s in both injection waveform conditions. When the waveform was constant, after one cardiac cycle, only blood flowed from the inlet under the condition that the dextran flow rate was 0 during the 1.0 s cardiac cycle. The 2.0 s set of these two cycles was simulated for 4.0 s. The result of only the second injection was evaluated. When the waveform was stepped, all the cases were simulated for three cycles (3.0 s). The result of the only third cycle was evaluated. A zero-pascal relative pressure condition was set at the extended outlet. The artery, catheter, and endoscope wall were defined as stationary, with a no-slip condition. The CFD calculations were carried out using a finite volume method in ANSYS CFX 2019 R1 (ANSYS Inc., Canonsburg, PA, USA).

### 2.3. Analysis Conditions

The density and viscosity of the blood were 1060 kg/m^3^ and 3.50 × 10^−3^ Pa s, respectively [25]. Additionally, the dextran’s density and viscosity were 1043 kg/m^3^ and 4.99 × 10^−3^ Pa s, respectively, which Otsuka Pharmaceutical Co., Ltd. cited. The blood and dextran flow was assumed to be that of an incompressible Newtonian fluid within a turbulent regime under transient conditions. The ***k-ε*** model was selected to model the turbulent flow with wall functions to model the mass fluxes in the turbulent boundary layer based on molar fluxes and molar fractions of the non-condensable fluid. The timestep size for the simulation was 0.005 s. The root mean square normalized residuals of the momentum and continuity of the iterative calculation were set to 0.0001.

### 2.4. Evaluation Indices

To evaluate the angioscopy visibility, a movie of the inside of the blood vessel from a camera viewpoint was produced using EnSight v.10.2.3 (Computational Engineering International Inc., Apex, NC, USA). During the movie’s production, the opaque volume due to the blood inside the blood vessel needed to be visualized. Then, the volume fraction of dextran (*Vf*) was used to define whether the volume was transparent or not. The *Vf* is the ratio of the volume of dextran to the total volume of the blood and dextran. The reason for choosing *Vf* as a parameter to define the opaque volume is that there is a relationship between the transparent mixture of the blood and dextran. The *Vf* was calculated by using Equation (1).
(1)$$Vf=\frac{V_{\mathrm{dextran}}}{V_{\mathrm{total}}}\times 100$$
where ***V_dextran_*** and ***V_total_*** represent the volume of the dextran and the total volume of the blood and dextran in one part, respectively. In this study, it was provisionally defined that the mixture of blood and dextran was transparent when *Vf* was more than 80%, referring to the optical density of fully oxygenated whole blood as a function of the hematocrit value in previous research [26].

From the movie of the inside of the blood vessel from the camera viewpoint, the visible area of the vessel wall for the camera within the camera view field (***CVA_visible_***) and the total area of the vessel wall within the camera view field (***CVA_total_***) could be calculated. When the function f(t) of the ***CVA_visible_*** percentage with respect to the ***CVA_total_*** was defined as *VAR* at a certain time t, *VAR* could be expressed as shown in Equation (2).
(2)$$VAR=\frac{CVA_{\mathrm{visible}}}{CVA_{\mathrm{total}}}\times 100$$

Additionally, the area ratio of the *VAR* (***AR_VAR_***) was defined to quantitatively evaluate the *VAR*. The ***AR_VAR_*** was the normalized integral of *VAR* during one cycle. The integral of 100% *VAR* during one cycle was defined as ***A_total_*** as expressed in Equation (3); ***AR_VAR_*** was calculated by using Equation (4).
(3)$$A_{\mathrm{total}}=CVA_{\mathrm{total}}\times \int_{one\;cycle}^{\;}\Delta t$$
(4)$$AR_{VAR}=\frac{\int_{one\;cycle}^{\;}VAR\;\Delta t}{A_{\mathrm{total}}}\times 100$$

## 3. Results

Figure 5 shows the *Vf* distribution, streamline, and normalized yz and zx velocity vector of the yz and zx cross-section plane at 0.100 s after starting to flush when the endoscope position was at the center, top, or bottom in the catheter. Figure 6 displays the *Vf* distribution, streamline, and normalized yz and zx velocity vector of the yz and zx cross-section plane at 0.100 s after flushing when the endoscope position was at the center or side in the catheter, and the injection waveform was stepped. Figure 7 shows the camera view field of each flushing condition at 0.100 s after starting to flush. The deep-red, light-blue, and green parts represent the invisible volume due to low *Vf* (<80%), the guidewire, and the blood-vessel wall, respectively. Figure 8 and Figure 9 indicate the *VAR* of each flushing condition as a function of time and ***AR_VAR_*** calculated from Figure 7, respectively.

### 3.1. Effect of Injection Waveform

As shown in Figure 5a,b, the dextran flowed straight and to the blood vessel’s top and bottom regions evenly. When the waveform was stepped, a dextran flow volume fraction at more than 80% covered the space between the camera and walls. Moreover, the flow reached the blood vessel’s top and bottom walls rather than a constant waveform. As shown in Figure 7a,b, the camera could capture a larger area of the blood-vessel wall when the waveform was stepped rather than constant. As can be observed in Figure 8a and Figure 9, the *VAR* for the stepped waveform was larger than that for the constant, and the ***AR_VAR_*** for the stepped waveform was also 8.5 times as high as that for the constant waveform.

### 3.2. Effect of Endoscope Position

From Figure 5c,d and Figure 6, it can be observed that the dextran flow went straight and in the direction in which the endoscope was contacting the inner wall of the catheter and reached the blood-vessel wall. These phenomena can be observed in Figure 7c–e. The camera could capture the wall in the direction in which the endoscope was contacting the catheter’s inner wall. Figure 8 shows that the *VAR* of the bottom (0°) model was larger than the *VAR*s for the other positions, including the position of the center model. In Figure 9, the ***AR_VAR_*** of the bottom (0°) model was larger than the ***AR_VAR_***s for the top, side, and center and around 1.9 times as high as that for the center.

### 3.3. Effect of Catheter Angle

As shown in Figure 5e–g, the dextran flow went straight and to the bottom region of the blood vessel. From Figure 7b,f–h, it can be observed that the camera could capture a larger area of the blood-vessel wall when the catheter angle was larger. In Figure 8c, it can be observed that the *VAR* of the bottom (3°) model was the largest and remained higher throughout the period of injection. From Figure 9, it can be observed that the ***AR_VAR_***s of the bottom (1°) and bottom (2°) models were higher than the ***AR_VAR_*** of the bottom (0°) model, but there was almost no difference in the ***AR_VAR_*** between the bottom (1°) and bottom (2°) models. Additionally, the ***AR_VAR_*** of the bottom (3°) model was the largest and around 1.5 times as high as that of the bottom (0°) model.

## 4. Discussion

Angioscopy is frequently used to observe the blood-vessel wall, and the conditions under which angioscopy is performed affects the success of operations. The flow of dextran depends on various flushing conditions, and the effects of each condition on the visibility are unclear. This study found out a way to realize clearer angioscopy under different values for the parameters of the injection waveform, endoscope position, and catheter angle, using CFD.

### 4.1. The Effect of Injection

Regarding the effect of the injection waveform, stepped injection can increase the visibility by 8.5 times compared to that realized with constant injection. This shows that increasing the volumetric ratio of dextran to blood increases the visibility. Additionally, this finding shows that the injection method during the operation can be improved while keeping the total volume of dextran the same.

### 4.2. The Effect of Angioscopy Shape on Visibility Improvement

The angle of the catheter and the position of the endoscope were analyzed to improve the angioscopy visibility. The effects of these geometry conditions are not even, and the position of the endoscope had a greater effect (1.9 times) than the catheter angle (1.5 times).

Overall, a three-degree angle for the catheter with stepped flow may improve the angioscopy visibility by 25 times compared with a zero-degree angle with constant injection. The combination of good flushing conditions will considerably improve the angioscopy visibility.

### 4.3. The Effects of the Parameters in Clinical Situations

The angle and position of the catheter depend on the position of the disease or the geometry of the artery in a given clinical situation. Adjusting the blood flow rate or dextran flow rate may offer potential for improving the *Vf*; however, the clinical situations frequently require the blood flow rate to be maintained. Thus, adjusting the injection waveform or injection timing can be tried as a method for improving the visibility.

### 4.4. Evaluation Indices

Here, we show the *VAR* and ***AR_VAR_*** for angioscopy visibility. The *VAR* represents the visibility at a given moment, and the ***AR_VAR_*** represents the total visibility over the period of injection. The *VAR* may be used for evaluating the visibility if a medical doctor needs to objectively evaluate a lesion. The ***AR_VAR_*** may be used for visibility evaluation if a medical doctor wants to visualize the whole region. This study proposes these quantitative evaluation indices as novel parameters for the assessment of angioscopic operations. These evaluation indices provide useful data that may be used to help clinicians in improving the efficacy of angioscopic procedures and to optimize device design through comparisons between models with different camera shapes, while their further application in clinical procedures needs to be investigated in more comprehensive studies.

### 4.5. Limitation

The ***k-ε*** model was selected to model the turbulent flow based on the preliminary experiment results and the previous studies. In the initial experiment, the flow during angioscopy was visualized using particle image velocimetry, and turbulent flow was observed around the endoscope despite the low Reynolds number. This turbulent flow is considered to have been caused by the complicated geometry of the camera. The ***k-ε*** model has been widely used to simulate turbulent blood flow [27,28,29]. The blood and dextran flows were assumed to represent those of an incompressible Newtonian fluid, which was an ideal condition. In addition, the blood vessel was assumed to be stationary in this study but possesses elasticity in a living body [30]. Therefore, comparison with experimental results will increase the practical relevance of this study; however, due to the complexity of the experimental study, we first performed this simulation to obtain preliminary results. In vitro experiments will be considered as further work on this topic. In clinical cases, some flushing conditions may not be used because of the patient’s geometrical condition. In this case, medical doctors may need to choose the best combination from these flushing conditions. For determining the ideal combination, an optimization method can be useful [31,32].

Additionally, idealized catheter and guidewire conditions were used in this study. The shapes of the catheter and guidewire have curves that may affect the visibility. Takashima et al. developed a simulator for the motion of a catheter and guidewire inside a blood vessel [33,34,35,36]. Simulating the flow under conditions reproducing the shape and motion of the catheter and guidewire in the blood vessels at a clinical site, with reference to those studies, will be considered for future work.

Additionally, the previous paper states that inducing too much pressure inside the coronary artery by flushing is unsafe for a patient [22]. The increase in pressure inside the coronary artery induced by dextran injection needs to be considered in the future.

This study concludes that the visibility depends on the temporal flush volume displacing the blood volume. However, in current clinical situations, several factors such as the blood volume and catheter position are not known before treatment. Experimental trials using an endoscope are necessary to confirm the mechanism of the effect on visibility.

To translate this simulation study to patient care, there is a need for preclinical (3D models) as well as clinical testing. Moreover, diseased walls of vessels are fragile. Therefore, a study considering different vessel thicknesses and fragility needs to be conducted before proceeding to real-life applications. Injecting dextran at a high flow rate could potentially rupture fragile vessels.

## 5. Conclusions

The purpose of this study was to investigate the effect of the injection waveform, endoscope position, and catheter angle on the angioscopy visibility by CFD analysis.

Regarding the injection waveform, it was found that the ***AR_VAR_*** for a stepped waveform was 8.5 times as high as that for a constant waveform. Regarding the endoscope position, the ***AR_VAR_*** of the bottom (0°) model was higher than the ***AR_VAR_***s of the top, side, and center models and around 1.9 times as high as that of the center model. Regarding the catheter angle, it turned out that the ***AR_VAR_***s of the bottom (1°) and bottom (2°) models were higher than the ***AR_VAR_*** of the bottom (0°) model, but there was almost no ***AR_VAR_*** difference between the bottom (1°) and bottom (2°) models. The ***AR_VAR_*** of the bottom (3°) model was the largest and around 1.5 times as high as that of the bottom (0°) model.

Therefore, it was found that the condition of the bottom (3°) model with a stepped waveform was the best for a higher ***AR_VAR_***. Additionally, the ***AR_VAR_*** of the bottom (3°) model with a stepped waveform was around 25 times as large as that of the bottom (0°) model with a constant waveform.

## Figures and Tables

**Figure 1 jfb-13-00069-f001:**
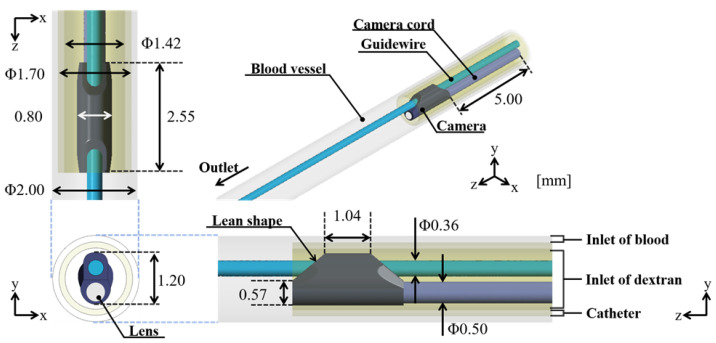
Basic model of blood vessel, catheter, and endoscope, called the center model. The straight catheter model was set in the center of the blood vessel. The endoscope was located at the center of the catheter.

**Figure 2 jfb-13-00069-f002:**
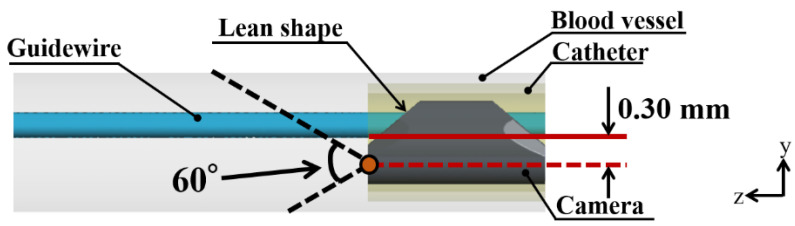
Camera viewing angle of the endoscope. The orange point and black dotted line represent the camera lens position and camera viewing angle, respectively. The red line denotes the center axis of the blood vessel. The red dashed line is passing through the camera lens position and perpendicular to the XY plane.

**Figure 3 jfb-13-00069-f003:**
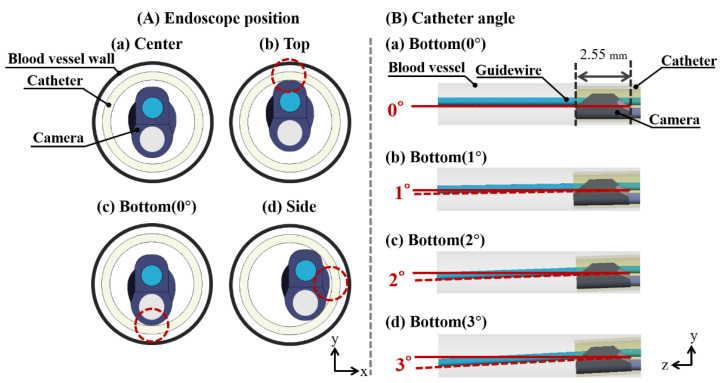
Models of different (**A**) endoscope positions in the catheter and (**B**) catheter angles. In (**A**), in addition to the model of (**a**) the center endoscope, the endoscope was brought into contact with the (**b**) top, (**c**) bottom, and (**d**) sidewall of the catheter. The red dashed circle of each position is the contact point between the endoscope and catheter wall. In (**B**), based on (**a**) the center model, the 2.55 mm-length part of the catheter tip was angled by 1, 2, and 3 degrees in the bottom direction. Red lines are the centerline of the blood vessel. Red dashed lines are the centerline of the angled part of the catheter.

**Figure 4 jfb-13-00069-f004:**
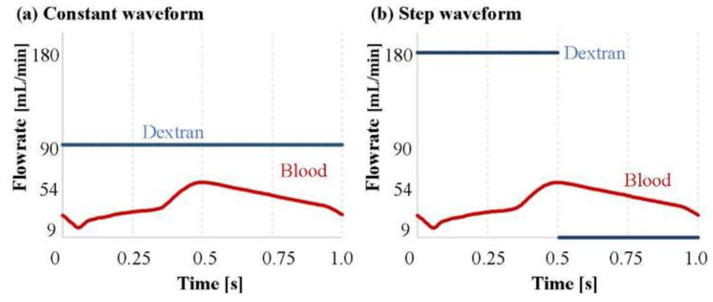
Inlet boundary condition of each injection waveform. The blood flow rate was the physiological pulsatile curve shown in both (**a**,**b**): (**a**) was continuous dextran injection under the constant-flow-rate condition; (**b**) was the dextran injection only during the 0.5 s systole. The injection when the blood flow rate was higher was stopped, and the maximum dextran flow rate was increased within the limitation of the dextran volume that could be injected.

**Figure 5 jfb-13-00069-f005:**
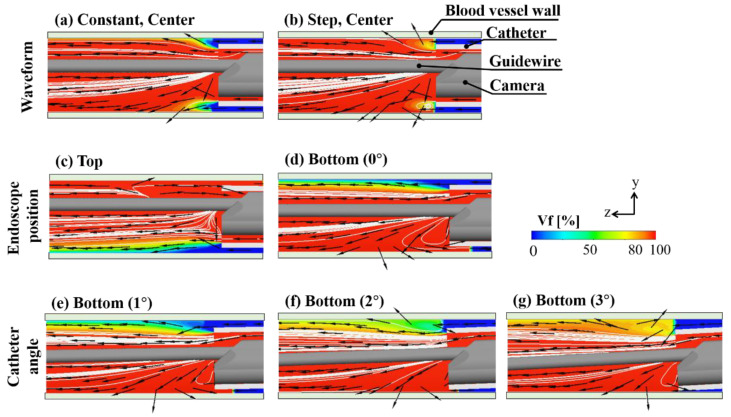
*Vf* distribution, streamline, and normalized yz velocity vector of yz cross-section plane at 0.100 s after starting to flush (x is the center of the blood vessel). Each row represents each flushing condition.

**Figure 6 jfb-13-00069-f006:**
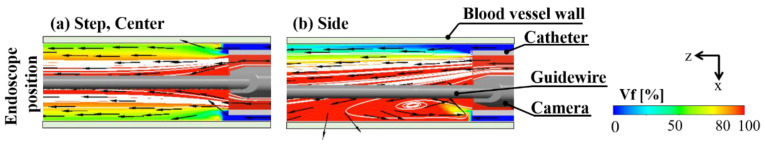
*Vf* distribution, streamline, and normalized yz velocity vector of zx cross-section at 0.100 s after starting to flush (y is the center of the blood vessel).

**Figure 7 jfb-13-00069-f007:**
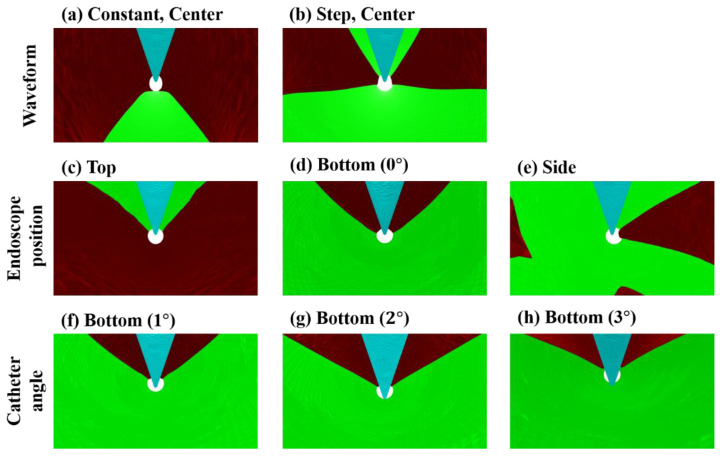
Camera view field of each flushing condition at 0.100 s after starting to flush. Each row represents each flushing condition. Deep red: the invisible volume due to low *Vf* (<80%). Light-blue: the guidewire. Green: the blood-vessel wall.

**Figure 8 jfb-13-00069-f008:**
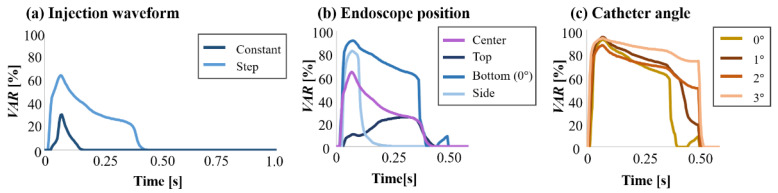
*VAR* evolution as a function of the injection waveform, the endoscope position, and the catheter angle.

**Figure 9 jfb-13-00069-f009:**
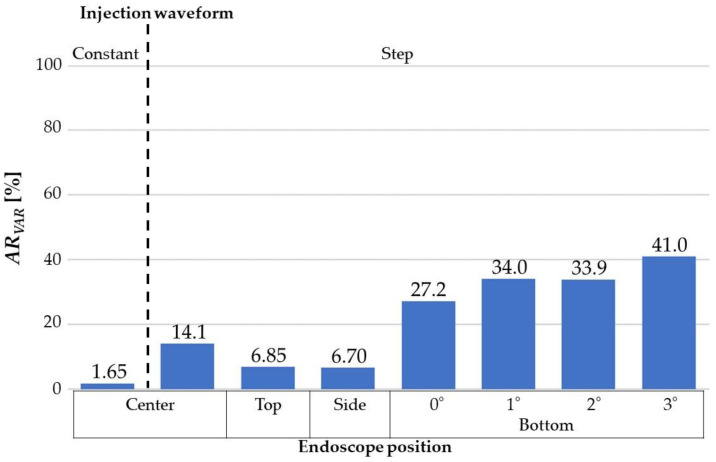
***AR_VAR_***s for all the flushing conditions.

**Table 1 jfb-13-00069-t001:** Flushing conditions considered to affect the angioscopy visibility. Underlined conditions were investigated in this study.

Flushing Conditions	
Geometry	Flow
Endoscope position in the catheter	Dextran injection waveform
Catheter angle	Dextran flow rate
Catheter position in the blood vessel	Dextran injection timing
Use of balloon	Blood flow rate

**Table 2 jfb-13-00069-t002:** The mesh numbers of each model.

Model	Mesh Number (Million)	Model	Mesh Number (Million)	Model	Mesh Number (Million)
center	3.3	Top	3.8	Bottom (1°)	6.0
		Bottom (0°)	5.6	Bottom (2°)	6.2
		Side	3.8	Bottom (3°)	5.7

## Data Availability

Not applicable.

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
