# Peer review of "A Parametric Study of Flushing Conditions for Improvement of Angioscopy Visibility"

_jfb, 2022, doi:10.3390/jfb13020069_

Round 1

Reviewer 1 Report

At the outset, I would like to congratulate the authors for conducting this research. The study aimed to explore conditions for a clearer angioscopy under different parameters of injection waveform, endoscope position, and catheter angle on the angioscopy visibility. 

The study demonstrated that the ARVAR with the set of step waveform, bottom endoscope, and three degrees angle catheter is the highest and around 25 times larger than that with a controlled set of the constant waveform, center endoscope, and 0 degrees. The study has merit and can be of interest to the readers. I have a few comments that need to be incorporated before the manuscript can be reconsidered.

Introduction: Please add your hypothesis. What did you hypothesize before conducting this research?

The methods and results sections are well-written. The limitations of the study are:

This study is based on software models. There is a need for preclinical (3D models) as well as clinical testing before the results can be translated to patient care.

Moreover, the diseased wall of the vessels would be fragile. So the study needs to be conducted on varying vessel thickness and fragility before real-life applications. The high flow rate of dextran could potentially rupture the fragile vessels. Please add these in the limitations of the manuscript.

The article also needs extensive English grammatical revision.

Author Response

Dear Reviewer,

We are thankful for your kind review and comments.

1. Introduction: Please add your hypothesis. What did you hypothesize before conducting this research?

->Thank you for your suggestion. We added a sentence as following;

"We hypothesize that the visibility can be improved not only with one condition, but with several conditions." We highlight the sentence. Please check it in page 2 (line 76-77).  

 2. The methods and results sections are well-written. The limitations of the study are:

This study is based on software models. There is a need for preclinical (3D models) as well as clinical testing before the results can be translated to patient care.

Moreover, the diseased wall of the vessels would be fragile. So the study needs to be conducted on varying vessel thickness and fragility before real-life applications. The high flow rate of dextran could potentially rupture the fragile vessels. Please add these in the limitations of the manuscript.

-> I am thankful for your good suggestion. We agree with your comment. We added it in the discussion with yellow lines. Please check them at page 10 (line 347-351).

3. The article also needs extensive English grammatical revision.

-> We asked the native to modify our English again. The English should be corrected.

Reviewer 2 Report

The manuscript by Mitsuzuka et al., titled: "A parametric study of flushing conditions for improvement of angioscopy visibility" is reporting on. novel approach in terms of better angioscopy visibility. The study has significant potential for clinical applications. 

The manuscript is very well written with excellent presentation, good structure and well organized. 

The reviewer would only recommend the inclusion of some statistics on the current state of the clinical practice as per the angioscopy status. 

Other than that very good work.

Author Response

Dear Reviewer,

1. The manuscript is very well written with excellent presentation, good structure and well organized. 

->I am thankful for your kind reviewing.

The reviewer would only recommend the inclusion of some statistics on the current state of the clinical practice as per the angioscopy status. 

-> I agree with your suggestion. I added two references and a sentence saying the statistical analyses. Please check the yellow line at page 2 (line 54-56). 
